# The Reactions of Orthodox Churches to Russia's Aggression towards Ukraine in the Light of the Postsecular Approach to IR Studies

Anna M. Solarz [1,*] and Iuliia Korniichuk [1,2,*]

1  Department of Regional and Global Studies, Faculty of Political Science and International Studies, University of Warsaw, 00-927 Warszawa, Poland
2  Department of Theology and Religious Studies, Faculty of History and Philosophy, National Pedagogical Drahomanov University, 01054 Kyiv, Ukraine
*  Correspondence: asolarz@uw.edu.pl (A.M.S.); i.korniichuk@uw.edu.pl (I.K.)

**Abstract:** Russia's war against Ukraine, in which the aggressor has been making use of religion, including theological rhetoric, to achieve its aims, has sparked reactions from Orthodox Churches all over the world. This has led to a revitalisation of social teaching, including discussions on war and peace within the Orthodox tradition. This may well become a further impetus for more in-depth research on religion and international relations, and possibly for more reappraisals of the secular identity of IR studies. An analysis of the attitudes of Orthodox Churches towards this war indicated that the authority of the Russian Orthodox Church, which considers itself the most important centre of Orthodox culture and civilisation, is waning. The reaction of other local churches showed that it is difficult to recognise the Russian Orthodox Church as such an authority. These revaluations may have a significant impact on Russia's place in the new international order, although much depends on the final outcome of the war it has started. We explain the different reactions of the churches, and we refer to the social teaching(s) on war of the Russian Church and the Ecumenical Patriarchate according to their official synodal documents. In this teaching, we can see two different approaches—Russian and Constantinopolitan. In the world of the Orthodox tradition, the former, whose practical expression was the atrocities committed during the ongoing war, seems to be rejected in favour of the latter, Constantinople. Finally, there is the question of how the reaction of the Orthodox Churches (analysed below), which have clashed with secularism in a different manner than the Western Churches, might contribute to the development of a postsecular awareness and, consequently, a postsecular identity for IR studies.

**Keywords:** religion; secularisation; postsecularism; international relations studies; Russia's war on Ukraine; Eastern Orthodox Church; social teaching of the Orthodox Church; war and peace



## 1. The 'Glass Ceiling' of the Secular Identity of IR Studies and the Challenges Posed by Russia's War against Ukraine

Studies on the role and place of religion in international relations, understood as the experience of international life, as well as scholarly reflections thereon (IR studies) have been increasing in number and depth for several decades, and especially since 9/11. This is attested by the growing number of publications (Brown 2020a, pp. 275–78), including textbooks for teaching and studying this issue, that collect and summarise what has already been revealed and which additionally introduce a good deal of scholarly reflection.[1] A connection between specific events with religious associations that have international repercussions and another wave of interest in this phenomenon in IR studies can be discerned in the literature (Bellin 2008). The authors of the present study believe that Russia's war against Ukraine, which has been ongoing since 2014, and which has moved up a gear

since the Russian invasion of 24 February 2022, is one such event. Both countries are majority Eastern Orthodox. As such, they draw on the same Christian traditions as the autocephalous Orthodox Churches of, e.g., Greece, Romania, and Serbia, and the Ecumenical Patriarchate of Constantinople, whose seat is in Istanbul. The war, in which Russia has been making use of religion, including theological rhetoric (Hovorun 2022c), to achieve its aims, has sparked reactions from Orthodox Churches all over the world. This has led to a revitalisation of social teaching, including discussions on war and peace within the Orthodox tradition. This may well become a further impetus for more in-depth research on religion and international relations, and possibly to more reappraisals of the secular identity of IR studies.

It is possible to obtain the impression that despite the development of studies of religion, for some time, IR scholars have been caught up in their own theoretical explanations. It is as if they have come up against a glass ceiling that they themselves installed for the sake of the 'scientificity' of their discipline (cf. Lindsay 2014; Solarz 2017). The authors of the present study contend that IR studies are confronted with the challenge of observing the intellectual structure that limits its development, a structure that is contingent on 'multiple regimes of knowledge and power', mainly of Western provenance (Brown 2020b, p. 296). This is relevant to the question of why interdisciplinary studies of religion in IR (e.g., studies that consider the theological aspect or religious studies in the broad meaning of the term) are still not able to break through to the mainstream remains. And this occurs despite the fact that it is becoming increasingly difficult to explain the international situation without factoring in religion, especially from a non-European perspective. Russia's war against Ukraine is evidence of this. This conflict cannot be analysed solely through the prism of Western theoretical conceptions that fail to consider the references to Orthodox theology and which evince no attempt to learn about this religion, as doing so would distort the image of the international setting in which the war is unfolding and could lead to erroneous conclusions.

The limitations of IR studies see" to 'ave to do with the 'scientificity' of the discipline being primarily understood as 'secularity'. This entails a deliberate lack of references to religion as an independent variable. Scott Thomas considers this failure to recognise religion in IR studies to be a natural consequence of the 'Westphalian presumption' (Thomas 2000), pursuant to which the modern international order, based on sovereign nations, has its roots in the Peace of Westphalia (1648), which—as the theory goes—ousted religion from the international domain. As Europe had endured many religious wars over a sustained period, religion was now seen to be extremely conducive to conflict, and as such, should be the preserve of national politics and free of outside intervention. Supposedly, this is how contemporary international relations developed with the omission of religion (Thomas 2000). The consequence of this for the development of IR was that it was deemed unnecessary to have religion included in any research studies. These studies instead came to be dominated by a secular academic narrative, which presupposed that 'Positioning the relationship between secularism and modernity as an axiomatic truth, this "Wesphalian presumption" is firmly situated within the study of International Relations as a condition of its possibility rather than an object of enquiry' (Thomas 2000). The problem is the 'Western-centricity' of this research approach, which is ineffectual in other cultural dimensions, including Islam and, more pertinent to this article, Eastern Orthodoxy.

There is a competing intellectual construction in IR studies, and it is based on a gradually emerging 'postsecular awareness'. Promoting this could be a reasonable way out of these difficulties (Mavelli and Petito 2012). This approach incorporates studies on religion into the IR mainstream and thereby provides better explanations, especially of the non-Western international actuality (Holmes 2015). As Holmes observes, we are witnessing the opening up of 'a methodological vista orientated toward rethinking the dialectical tensions between religion and secular notions of modernity' (Cf. Mavelli and Petito 2012). The late-Westphalian IR development stage is favourable for promoting the postsecular approach. At present, this stage is associated with the decline in the liberal order, whose

expression and possibly finishing touch is Russian aggression cloaked in quasireligious garb and authorised by the most senior Russian Orthodox clergy.

The analysis of the attitudes of Orthodox Churches towards this war (presented below) indicate that the authority of the Russian Orthodox Church, which considers itself the most important centre of Orthodox culture and civilisation (Панарин 2014), is waning.[2] The reactions of other local Churches show how difficult it is to ascribe such authority to the Russian Orthodox Church. This could have a major bearing on Russia's place in the new international order, although a great deal hinges on the final outcome of the war that it has launched. While attempting to elucidate the attitudes of the Churches, the present authors attempt to differentiate the teachings of the Russian Orthodox Church and the Ecumenical Patriarchate on the subject of war according to their official synodal documents. Two incipient approaches, viz. the Russian and the Constantinopolitan, can be seen in this teaching. Although part of the same tradition, these Churches are characterised by different sensibilities, as is evident in, e.g., the language they use and different emphases on various aspects. Moreover, in the Orthodox tradition, the former, whose practical expression has unfortunately become the cruelties perpetrated during the ongoing war, seems to be being rejected in favour of the latter. Finally, there is the question of how the reaction of the Orthodox Churches, analysed below, which have clashed with secularism in a different manner than the Western Churches, might contribute to the development of a postsecular awareness and, consequently, a postsecular identity for IR studies.

## 2. Orthodox Churches against Russian Aggression in Ukraine

It is worth emphasising that since Russia's full-scale invasion (February 2022), every single local Orthodox Church, without exception, in one way or another has reacted to this aggression with a Synod resolution, statements from senior clergy, or lay initiatives. The Orthodox theologian Demacopoulos divides the reactions of the Orthodox Churches regarding the next stage of war (since 2022) into four groups: the ridiculous, the generic, the strident, and the surprising (Demacopoulos 2022).[3]

To the first group, Demacopoulos attributed the statements of the head of the Russian Orthodox Church, who supported the invasion (or, in Patriarch Kirill's words, 'the current events'). During the eight years of the ongoing war, the position of the Moscow Patriarchate on war has gone through several twists and turns that can be arbitrarily divided into three stages. At first (2014–2015), the church was working on developing an alternative narrative story and promoting it on the international stage. The elements of this story were the presentation of the war as an internal civil and interfaith conflict ('a fratricidal civil war' (His Holiness Patriarch Kirill Calls Primates of Local Orthodox Churches to Raise Their Voice in Defence of Orthodox Christians in the East of Ukraine 2014) and 'the attempts of the Uniates and schismatics to do harm to the canonical Orthodoxy' (Primate of Russian Orthodox Church Sends Letter to United Nations, Council of Europe, and OSCE Concerning Persecution of Ukrainian Orthodox Church in the Situation of Armed Conflict in the South-East of Ukraine 2014)). Additionally, they carefully avoided mentioning the role of Russian troops or violations of human rights on the occupied territories, including violations in the name of 'protecting Russian Orthodoxy' (activities of the Russian Orthodox army and neo-Cossack formations (Darczewska 2017)), repression against non-Orthodox churches, and support of the occupational forces by individual priests. At the same time, the patriarch and the higher clergy avoided visiting occupied territories and even temporarily limited the rhetoric regarding the 'Russkij Mir'—the doctrine that became the ideological basis of the aggression.[4] Such a hybrid approach contributed to the fact that a number of researchers investigating the role of religion in the Russian–Ukrainian war, although recognising the Russian Orthodox Church as an instrument of Russian politics, interpreted the role of the church as neutral.[5]

The starting point of the second stage was the request for autocephaly for the Orthodox community in Ukraine. A new round of negotiations on granting it began in the summer of 2015 and became public in 2018. The main argument for granting autocephaly

was the mass exodus of faithful from the Ukrainian Orthodox Church (of the Moscow Patriarchate) due to the position of the Moscow Patriarchate regarding the war. The faithful faced a difficult moral dilemma, choosing between staying in a church that does not oppose aggression against their own country, or joining the alternative Ukrainian Orthodox Churches whose status is not recognised by the Ecumenical Orthodoxy. Despite the status of the largest religious organisation in Ukraine (more than 1/3 of registered religious communities), the level of self-identification with UOC(MP) had fallen to approximately 13% (Bogdan 2016) by 2016 (shortly before the establishment of the OCU) and later reached an all-time low of 4% (Dynamics of Religious Self-Identification of the Population of Ukraine: Results of a Telephone Survey Conducted on July 6–20 2022) in 2022 (after the full-scale invasion).

So far, the autocephaly of the unified Orthodox Church of Ukraine, which was granted in January 2019, has been recognised by four local churches: the Constantinople and Alexandrian patriarchates and the Churches of Greece and Cyprus. Such a division in recognition once again revealed the old alliances within the Orthodox Church, as pointed out by researchers (Curanović 2007; Leustean 2018). At the same time, this case became an illustration of the limits of such alliances. After the creation and recognition of the OCU, the ROC completely stopped Eucharistic communication with Constantinople and Alexandria and selectively communicated with the Churches of Greece and Cyprus. This measure is the highest degree of institutional sanctions of the Orthodox Church. Despite the differences in the assessment of Ukrainian autocephaly, none of the local Orthodox Churches followed the Russian example, even Russia's traditionally close allies such as the Serbian Church.

The granting of autocephaly became visible, but it was not the only manifestation of the effect of the war on the administrative structure of the church. A similar process, albeit on a smaller scale, began after the full-scale invasion (24 February 2022), and consisted of the third stage of the development of attitudes towards the war. In the early days of the full-scale war, the Moscow Patriarch not only supported the Russian invasion, but also defined the war as a higher 'metaphysical' struggle against 'so-called values, that are offered today by those who claim world power,' the most visible sign of which are gay parades (Патриаршая Проповедь в Неделю Сыропустную После Литургии в Храме Христа Спасителя [Patriarchal Sermon on Cheesefare Week after the Liturgy at the Cathedral of Christ the Saviour; 6 March 2022] 2022). In later months, this turned into a declaration of washing away the sins of the Russian soldiers who died in Ukraine (Moscow Patriarch: Russian War Dead Have Their Sins Forgiven 2022), which is a de facto declaration of the war as holy. Not all ROC parishes outside Russia shared their support for the war. Already in the initial weeks, some of those communities announced their break with the Moscow Patriarchate[6] or criticised the position of the church and demanded greater autonomy (Lithuanian Orthodox Church Condemns Russia's War on Ukraine [18 March 2022] 2022).

Such demands went the farthest in the Baltic states, whose governments gradually started to advocate the independence of Orthodox communities on their own territories in order to reduce the influence of the Russian Orthodox Church. In addition to the already existing parallel jurisdictions of Moscow and Constantinople in Estonia, which caused the largest conflict between Orthodox churches before Ukraine, the governments of Lithuania and Latvia directed similar requests to religious leaders in Moscow and Constantinople for autocephaly (Kuczyńska-Zonik 2022).

The next two groups of churches Demacopoulos identified as the generic (Bulgarian (Обръщение на Негово Светейшество Българския патриарх Неофит за мир по повод военните действия в Украйна [Address of His Holiness the Bulgarian Patriarch Neophyte for Peace on the Occasion of the Military Actions in Ukraine; 25 February 2022] 2022), Serbian (Патријарх Порфирије: Посведочимо Јеванђељску и Хришћанску Љубав Према Страдалној Браћи у Украјини [Patriarch Porphyry: Let Us Bear Witness to the Gospel and Christian Love for the Suffering Brothers in Ukraine; 6 March 2022] 2022), Jerusalem (Statement on Situation in Ukraine by His Beatitude Patriarch of Jerusalem Theophilos III

[27 February 2022] 2022), Georgian (სრულიად საქართველოს კათოლიკოს-პატრიარქის განცხადება (24 February 2022) [Statement of the Catholicos-Patriarch of All Georgia (24 February 2022)] 2022 Churches), and the strident (Ecumenical Patriarchate (Statement by His All-Holiness Ecumenical Patriarch Bartholomew about the War in Ukraine (Sunday, 27 February 2022) 2022), Greek (Περί Τοῦ Πολέμου Στήν Οὐκρανία [On the War in Ukraine (Encyclical Letter of the Holy Synod of the Church of Greece; 17 March 2022)] 2022), and Romanian (Iftimiu 2022) Churches). The main difference between them is the expressiveness of the rhetoric and the readiness/unwillingness not just to call for peace but to call the war a war and give it an assessment (indicate the parties involved, their roles, and determine the nature of the conflict as aggressive or defensive).

It should be noted that since the publication of Demacopoulos' article, all Orthodox Churches have made official statements about this war. The vocabulary and evaluations of some of them have undergone significant changes. An illustrative example is the Georgian Orthodox Church, which in the first days of war limited itself to just a cautious statement in support of peace. Later, in addition to the expected involvement of religious organisations in humanitarian aid and the problems of displaced people, the church condemned the atrocities 'at the site of combat operations of Russian troops in Bucha' (საქართველოს საპატრიარქოს განცხადება 5 April 2022 [Statement of the Patriarchate of Georgia 5 April 2022] 2022) and offered condolences to the families of fallen Georgian volunteer soldiers of the Ukrainian Foreign Legion.[7] The Albanian Church, for example, even issued a separate statement regarding the inaccuracy of the claim that the Church was avoiding any mention of Russia as an aggressor state (Inaccurate Claim Regarding the Statements of Archbishop Anastasios [13 May 2022] 2022).

On the one hand, in general, the churches condemned war as a tool for solving problems in these documents ('No provocation, no pursuit and no pretext can justify the atrocity of war < . . . > Not only does war not solve the problems, but it feeds the circle of violence, hatred, pain, uprooting, refugees, hunger and loss of human life itself, which as Christians we must respect, protect and honour' (Περί Τοῦ Πολέμου Στήν Οὐκρανία [On the War in Ukraine (Encyclical Letter of the Holy Synod of the Church of Greece; 17 March 2022)] 2022). On the other hand, they emphasised the illegitimacy of the military operations, described using the epithets as 'unjust' (Message of Support from His Beatitude Theodoros II, Pope and Patriarch of Alexandria and All Africa to the People of the Ukraine [1 March 2022] 2022), 'unprovoked', 'beyond every sense of law and morality' (Statement by His All-Holiness Ecumenical Patriarch Bartholomew about the War in Ukraine (Sunday, 27 February 2022) 2022), 'wicked and incomprehensible' (Uchwała Soboru Biskupów Ws. Wojny w Ukrainie [Resolution of the Council of Bishops on the War in Ukraine; 22 March 2022] 2022), etc. Some churches even went further, appealing to Russian authorities to stop the war, and these demands remain unheard so far.

In the end, Demacopoulos attributes the position of the Ukrainian Orthodox Church (of the Moscow Patriarchate) to the fourth group (the surprising), as the head condemned the war on 24 February and called on the Russian president to end it. Since 2014, the statements and gestures of the UOC(MP) have been rather consistent with the Russian Orthodox Church, which has caused a misunderstanding between laity and the latter's mass refusal to self-identify with this church. Despite the long opposition to the movement for autocephaly, with the onset of open, full-scale war, the church declared its 'full self-determination and independence' in May 2022 (Постанова Собору Української Православної Церкви від 27 травня 2022 року [Resolution of the Council of the Ukrainian Orthodox Church Dated 27 May 2022] 2022, Article 4). Even months after this decision was made, it is still hard to assess whether the separation is real—it remains unclear whether its hierarchs still participate in the Synod of the ROC, or whether they have notified other local churches about their new status, etc. The only clear consequence of the decision of 27 May was the de facto annexation (taking direct control) of the UOC(MP)'s communities on the occupied territories by the Moscow Patriarchate (Журналы Священного Синода От 25 Августа 2022 Года [Journals of the Holy Synod from 25 August 2022] 2022, No. 63).

In January 2023, the clergy and faithful of the UOC(MP) issued an open and public request to the church leaders to provide a clear explanation regarding the status of the church, its connection to Moscow, its assessment of collaborations with the occupation forces among the clergy, and the facts about the annexation of Orthodox communities on the occupied territories (Звернення Духовенства Та Вірян УПЦ До Священного Синоду Та Єпископату УПЦ [Address of the Clergy and Faithful of the UOC to the Holy Synod and the Episcopate of the UOC] 2023).

As can be seen, most Orthodox Churches have tended to censure this aggression unleashed in the Orthodox world. Nor is there any acceptance, let alone support, for the position of the Russian Orthodox Church. This is due to pressure from the faithful and the influence of public opinion, which is opposed to the war and its religious legitimisation. The pro-Russian stance of the hierarchy has become unacceptable in these circumstances. The letter to the faithful from the Primate of the autocephalic Polish Orthodox Church, Metropolitan Sawa of Warsaw and all Poland, dated 4 February 2023, is a case in point. In it, Sawa apologises for a 'customary personal despatch' he had sent few days before to Patriarch Kirill of Moscow and all Rus' on the occasion of the 14th anniversary of the latter's enthronement. The letter, which the patriarchate published, together with congratulations from, inter alia, Putin and Lukashenko, could be construed as supporting the Russian Orthodox Church's position on the war.[8] In his letter of apology, Sawa strongly distanced himself from the Russian aggression and reiterated that he had condemned it from the outset, referring to it as the 'Criminal Invasion of the Russian Federation'. He went on to emphasise that he had repeatedly expressed his 'astonishment and embarrassment at what Kirill had been saying about the war in Ukraine' and that he had not and did not share these views which, in his opinion, 'did not serve to restore peace to Ukraine' and 'had a negative influence on the perception of Eastern Orthodoxy'.[9] The letter also emphasised the Metropolitan's unwavering support for the need for the Orthodox Church in Ukraine to become independent and entreated the faithful and all Poles to forgive him for the error of his imprudent customary despatch, which was sent under different geopolitical circumstances than those which currently prevailed (Sawa 2023).

### 3. The Shaping of Social Concepts and Traditional Orthodox Attitudes towards the War

Before attempting to explain the differences in the reactions to Russian aggression mentioned above, it is worth taking a look at Orthodox social teaching(s), as it includes a reflection on war. It should be emphasised that because Orthodox Churches have historically been connected with state power, they have not always been inclined, or sometimes even able, to articulate political events. Constructing a body of systematised social teaching, which would reflect the Church's position on social issues, is a relatively new development. The most significant attempt to generalise the Orthodox social position was made by the Holy and Great Council of the Orthodox Church, held in Crete in 2016. The Council adopted several documents that marked the general contours of understanding the role of the Orthodox Church in the modern world and marked the beginning of a renewal of Orthodox social teaching. However, a few local churches refused to participate in the event.[10] This was a manifestation of existing disputes between local churches, and it made it difficult to generalise the Orthodox position on certain sociopolitical issues, including war and peace.

#### 3.1. The Basis of the Social Concept of the Russian Orthodox Church

The Russian Orthodox Church was one of the first to try to formulate the general outlines of the social teaching of the Orthodox Church. In 2000, it adopted a document titled *The Basis of the Social Concept of the Russian Orthodox Church* (BSC) (Основы социальной концепции Русской Православной Церкви [Bases of the Social Concept of the Russian Orthodox Church] 2000). The document was designed to be open ended and was expected to be gradually extended with separate documents dedicated to certain aspects of social

teaching. The issue of war and peace is one of over a dozen problems[11] raised in the BSC (Chapter VIII).

From the very beginning of the chapter on war and peace, the document clearly condemns war and murder as evil (chp. VIII.1)[12]. At the same time, the Russian Orthodox Church does not forbid its faithful from 'participating in hostilities if at stake is the security of their neighbours and the restoration of trampled justice' (chp. VIII.2).

The document does not provide the criteria for a protective/defensive war or the conditions that justify it, limiting itself to mentioning the criteria of a just war developed by Augustine the Blessed 'in the Western Christian tradition.' Instead of specifying the understanding of the conditions for a justifiable war, the document turns to the broader issue of 'moral truth' in international relations, which is presented blurrily but in close connection with the issues of civic responsibility and the state. The church assesses the morality of the invasion, according to the principles of 'love one's neighbours, *[one's] people* and *Fatherland*; *understanding of the needs of other nations*; conviction that it is impossible to serve one's country through immoral means' (italics added) (chp. VIII.3). The authors of the document avoid any further specification or interpretation of these principles.

As if realising the possible ambiguity of the interpretations of these principles and the problematic nature of using them in real life, the document acknowledges the difficulty of distinguishing between aggressive and defensive actions in the modern world. That is why the church is entrusted with assessing the justifiability and morality of military actions in each specific case. It is interesting that according to the document, such an assessment is considered necessary not only in the situation of the outbreak of war but also when there is a very threat of military operations commencing. 'In the present system of international relations, it is sometimes difficult to distinguish an aggressive war from a defensive war. The distinction between the two is especially subtle where one or two states, or the world community, initiate hostilities on the grounds that it is necessary to protect the people who fell victim to aggression (see chp. XV.1). In this regard, the question of whether the Church should support or deplore the hostilities needs to be given special consideration every time hostilities are initiated or threaten to begin' (chp. VIII.3).

One of the signs of a justifiable war is the methods through which the war is waged and the treatment of prisoners and civilians of the opposite side, who must be treated with respect. In participating in military actions, one is expected to dissociate himself from evil ('for in struggling with sin it is important to avoid sharing in it'), which results in the controversial justification of taking a human life: 'The Christian moral law deplores not the struggle with sin, not the use of force towards its bearer and not even taking another's life in the last resort, but rather malice in the human heart and the desire to humiliate or destroy whosoever it may be'. In this context, a special role is assigned to the priesthood, which ensures the support of 'the established Orthodox traditions of service to the fatherland' (chp. VIII.4).

Sooy points to several controversial aspects of the BSC's interpretation of the issue of war (Sooy 2018). First, the document basically introduces the just war theory (JWT), which is alien to the Eastern Orthodox tradition. This attempt becomes even more direct in the draft of the *Catechism of the ROC*. Another piece of the same problem is the kindred and half-hearted adaption of the JWT. Without the teaching of jus post bellum, the very understanding of a just war shifts from the historical attempt of limiting war to creating the grounds for starting one (Sooy 2018, pp. 55–56). As Kyrou and Prodromou show, even though historically, theologians in Byzantium were familiar with the concepts of a just and holy war,[13] and they deliberately rejected it as being both dangerously challenging and unacceptable for the Orthodox tradition (Kyrou and Prodromou 2017, p. 226).[14]

Second, an inherently contradictory aspect is the attempt to justify murder as morally acceptable if it was committed without 'malice in the human heart' (chp. VIII.1). This statement not only directly contradicts the previously stated unacceptability of murder (chp. VIII.1), but it also leaves a great deal of unlimited room for interpretation.[15] Third, the logical sequence of presenting the argument in the chapter creates a false impression

that there is a special spiritual reward for warriors (Sooy 2018, pp. 51–52). The document stresses the great number of warriors canonised by the Orthodox church (chp. VIII.2) but does not clearly mention that they were not canonised for their military successes. This presentation might be interpreted as a step toward understanding war as holy. In his later sermons and speeches, Patriarch Kirill came even closer to the interpretation of the war against Ukraine as holy, proclaiming that the Russian soldiers who died in Ukraine will receive forgiveness for their sins (Moscow Patriarch: Russian War Dead Have Their Sins Forgiven 2022).

To the arguments given by Sooy, it is also worth adding the ambiguity and vagueness of the principles of 'understanding the needs of other peoples' or the concordance of love for one's 'people and Fatherland' with Christian universalism or transnationalism, with which the Russian Orthodox Church positions itself.

In general, the document carefully avoids any specification or development of problematic points, which complicates its practical application. On the other hand, the general presentation allowed the document to become a starting point for dialogue on social issues, not only with other Christian traditions (such as the Oriental or Protestant churches) but also with other religions (Judaism and Islam). With minor corrections and changes, his text became the basis for a number of 'social concepts' of other religious organisations in the Russian Federation, which were adapted in the years that followed.[16]

*3.2. 'For the Life of the World': Ecumenical Patriarchate*

In 2020, the Ecumenical Patriarchate endorsed the social document 'For the Life of the World: Toward a Social Ethos of the Orthodox Church' (2020). The document was written by a group of Orthodox theologians from all over the world at the request of Patriarch Bartholomew. As Hovorun and Chryssavgis show, the document is a product of a broader dialogue and is based on the ideas and works of previous generations of theologians, reflections on developments in political theology in other Christian denominations, and the elaborations of the Holy and Great Council of Crete (2016) (Hovorun 2022b; Chryssavgis 2020).

In contrast to BSC, the authors of FLW set themselves the task of creating a document that would be more inclusive and 'open to the world'; one that would become 'an initiation of a continuing conversation'. This intention is reflected in the structure of the document, which is more focused on modern social problems (poverty, refugees, slavery, racism, etc.) and less on the state and is more compassionate than commanding in tone. Among the more than thirty thematic blocks, FLW also raises the issues of secularisation, justice, violence, war, and peace.

As in the previous document (BSC), the authors distinguish between the 'violence of nature' (because of alienation from God) and 'the violence intentionally perpetrated by rational human agents' (chp. V, §42). War, which is the latter organised mass scale, is 'the most terrible manifestation of the reign of sin and death in all things': 'Nothing is more contrary to God's will for creatures fashioned in his image and likeness than violence one against another, and nothing more sacrilegious than the organized practice of mass killing. All human violence is in some sense a rebellion against God and the divinely created order' (chp. V, §42).

Condemning all forms of violence as 'the intentional use of physical, psychological, fiscal, or social force against others or against oneself' (chp. V, §43), the Church 'recognizes the *tragic necessity* of individuals or communities or states using force to defend themselves and others from the immediate threat of violence' (chp. V, §45) (emphasis added). In this situation, a great deal of responsibility lies with legitimate governments, which have to make every effort to establish 'just and compassionate laws' and 'grant equal protection and liberty to all the communities' (including ethnic and religious ones) to ensure a peaceful coexistence. It is additionally recognised that there are cases where the preservation of peace is impossible. In such cases, 'self-defence without spite' and a judicious, nonexces-

sive use of force in combination with a 'sincere effort to bring about reconciliation, forgiveness, and healing' is considered justifiable (chp. V, §45).

The Orthodox Church does not insist on 'a strictly pacifist response to war' or prohibit the faithful from 'serving in the military or the police force'. At the same time, it rejects any theory of 'just' or 'holy' war. This is because there may be hidden motives behind such wars, e.g., 'hatred, racism, revenge, selfishness, economic exploitation, nationalism or the quest for personal glory'. Even though the Orthodox tradition rejects the concept of a 'just' war, it uses the two criteria laid down by Augustine, viz. war as a last resort and the proportional use of force. This establishes a common ground for dialogue with the Catholic Church. Another point of contact could be the reference in the document to the threats posed by modern military technologies (chp. V, §46).

Some of the articles in the document specifically consider the state's role in organised violence and its prevention. The Church condemns the fixation of states on war, its neglect of social needs in favour of excessive armament, 'military expansion', and the strengthening of the 'military-industrial complex' (chp. V, §38). It emphasises the duty of Christians and Churches to respond to injustice and the responsibility of citizens for the violence committed by their state: 'even if [violence] prosecuted by the state on our behalf without our awareness—we are to some degree complicit in the sin of Cain' (chp. IV, §38; chp. V, §43).

The document condemns terrorist acts and the use of terror against the civilian population. Deliberate killings of civilians cannot be justified under any circumstances, including political ideas or tasks (chp. V, §46, partially chp. IV, §38, chp. V, §43).

Local Orthodox churches, however, differ in their interpretations of issues related to war. While they all condemn war and murder, they place different emphases on the causes and permissibility/justification of war.[17]

## 4. Attitudes of Orthodox Churches towards War, and Postsecularism in IR Studies. The Reciprocally Mediated Hermeneutical Matrices of Miłosz Puczydłowski

The attitudes of the Orthodox Churches towards war, both in terms of the positions they take (as analysed by Demacopoulos) and in their social teaching on war, reveal certain differences in their approach. These are discernible in the understated language used and their differently distributed emphases. Nevertheless, Russia's war against Ukraine has become a striking example of the practical influence of religion on politics and of how religious concepts can be used as a foundation on which to legitimise international military aggression.[18] The war can also be said to have had the opposite effect in that it has profoundly affected the interior life of the Church—from the interpretation and application of its social teaching(s) to its administration reorganisation (changing the status of local churches and the jurisdictions of particular parishes). The limited influence of religious organisations should also be noted. Despite their condemnations of the war and their repeated appeals to the secular authorities, religious leaders have been ineffectual in preventing or stopping it.

It does not follow, however, that the position of the Churches will not have any long-term influence on political relations, especially with countries that are culturally close. Recognising the religious dimension of international relations (which here is the positions of the various Orthodox churches and the relations between them in the face of a war between two countries belonging to the same religious tradition) renders the events and processes accompanying that war more comprehensible and more easily explicable. Proponents of a 'postsecular turn' might query the justification of this position, which is one of many 'epistemic turns' that have been made in the social sciences and the humanities in recent decades. However, although this turn (as with many others) occurs within the Western scholarly and cultural milieu, it can be used to study the social reality of other areas, including those that remain within the scope of Orthodox culture. The postsecular approach has grown out of critical reflection on the state of Western scholarship, which is seen as closed to different perspectives. One of the stated aims of postsecularism has therefore been to open it up. This is also the task that this article has set itself, hence the

examination of the roots and main ideas of postsecularism below. Postsecular applications are also increasingly found in the study of the international state of affairs. This has endowed IR studies with a whole new complexion. The postsecular approach proposed in the present article is based on the deliberations and reflections of the Polish philosopher Miłosz Puczydłowski. Together with the investigations of other researchers, these can build a foundation for the development of a postsecular identity for IR studies.

Applying this analytical approach to international relations to an area dominated by the Eastern Orthodox tradition requires additional reflections concerning the growth of secularism within this tradition. A separate section is therefore devoted to this issue. The present article treats this point as a step in the development of this postsecular identity as it explains the cultural nuances of the non-Western regions of the world and thereby accommodates the non-Western research perspective postulated in the 'postsecular turn'. The usefulness of the postsecular category is affirmed by European and American researchers whose worldviews vary from liberal (e.g., Rawls) to neo-Marxist (e.g., Žižek and Badiou) and is being referenced by researchers from other cultural milieux as well. *The Routledge Handbook of Postsecularity* is one of the volumes on this topic compiled in recent years. The main initiator of this undertaking, Justin Beaumont, has stressed that postsecularism opens a space for intellectual and political engagement that is radically pluralistic and open to dialogue (cf. Obirek 2021, p. 90). For his part, the Turkish anthropologist Umut Parmaksiz has observed that postsecularism poses the challenge of having truth (whether religious or not) included in a debate without calling the open nature of that debate into question (cf. Obirek 2021, pp. 89–90).

According to what has been stated above, this challenge has a special place in IR studies on account of the 'Westphalian presumption'. Our analysis of the stance of the Orthodox Churches towards Russia's war in Ukraine unequivocally convinces us that incorporating the religious dimension into our reflections on international relations can help make specific international events more comprehensible and explicable. It also gives greater latitude in anticipating the unfolding of events, e.g., by showing that the waning of the Russian Orthodox Church's authority will lead to the waning of Moscow's authority among Orthodox countries.[19] Tension between the politicosecular (the war) and religious (moral reflections on good and evil) spheres and their interaction and reciprocal conditioning, which do not imply direct opposition, can be observed in the processes analysed above.

Something approaching an explanation of the developments observed by postsecular scholars can be found in *Geographies of Postsecularity: Re-envisioning Politics, Subjectivity and Ethics* (2019). Although this work stresses that 'the religious and the secular are often defined as binary opposites', it also explores 'alternative configurations of these terms' (Cloke et al. 2019). The postsecular approach essentially involves observing the mutual connections and dependencies between the secular and the religious. Although such scholars as Talal Assad, José Casanova, Charles Taylor, and Peter Berger have sought to trace the sources of postsecular thought, the thinking of Jürgen Habermas, which was a response to 9/11, appears to be especially significant (cf. Barbato 2020). Card. Joseph Ratzinger agreed with the gist of Habermas's postsecular philosophy, stressing that there was a need for mutual learning and mutual purification through secular and religious thought (cf. Obirek 2021, p. 92). The discussion between these two thinkers was in itself postsecular, as one of the disputants was a secular atheist philosopher and the other a devout Catholic theologian. The commonalities in their views and convictions became an inspiration for the development of postsecular thought.

This postsecular thinking crossed over into IR as well in the 2010s (Mavelli and Petito 2012; Mavelli and Wilson 2016; Barbato and Kratochwil 2009). IR research recognised that the social sciences employ the term 'postsecularity' in two related, albeit different, ways. The first is more descriptive, i.e., 'it has been used to explain the return or resilience of religious traditions in modern life' (Mavelli and Petito 2012, p. 931). Mavelli and Petito contend that, on the one hand, this contributes to the development of 'conceptual frameworks that could account for this unexpected feature of modernity beyond the paradig-

matic assumptions of the secularisation theory', and on the other, constitutes a 'plea for new models of politics able to include religious views'. The second way is more normative, i.e., 'the postsecular has emerged as a form of radical theorising and critique prompted by the idea that values such as democracy, freedom, equality, inclusion, and justice may not necessarily be best pursued within an exclusively immanent secular framework. Quite the opposite, the secular may well be a potential site of isolation, domination, violence, and exclusion' (Mavelli and Petito 2012). By drawing on the considerations of various scholars, Mavelli and Petito noticed that this approach shed light on many aspects of IR studies. In particular, this concerns: (i) 'the centrality of the secular as a modern epistemic category'; (ii) 'secularism as a tool of power of the modern state'; (iii) 'the Eurocentric matrix of secularism and its powerful working in the postcolonial world'; (iv) 'how the secular is often constructed and reproduced against the ultimate "Other" of Islam'; (v) 'the limits of secular instrumental reason and the necessity of recovering the moral intuitions of faith as a necessary component of modernity'; and (last but not least) (vi) 'the articulation of the secular in non-Western political traditions and as part of a global civilisational dialogue' (Mavelli and Petito 2012). They summarise these considerations as: 'Moving from different sensibilities and concerns, these perspectives articulate sketches of postsecular visions that encourage us to think beyond current secular framework' (Mavelli and Petito 2012). We find that we applied the same approach to analysing the position of the Orthodox Churches regarding Russia's ongoing war against Ukraine. The war, viewed as having no religious references whatsoever, has a completely different nature than most conflicts.

To return to the Russian aggression in Ukraine and the Ecumenical Orthodoxy's reaction to it, however, Puczydłowski has other thoughts on postsecularism that are worthy of examination. His approach is close to the Habermas–Ratzinger discussion and may prove to be a useful research tool in IR studies as well. It consists of a certain method of understanding interconnected phenomena and events that have both religious and secular aspects. This raises the question of how to translate a postsecular reappraisal of the social science into a practical analysis of international events. Those conducting IR studies have a lot of ideas and advice on this, but not too many practical examples (e.g., Barbato 2020). Every comment on applying postsecularism in IR studies, however, should be given due consideration. For example, E. Shakman Hurd is opposed to any 'restorative narrative' that downplays the 'violent face' of religion and highlights its positive aspects. She is consequently against whitewashing the role of religion by way of compensating for its long absence in IR studies. It is hard not to concur with this approach. The two faces of religion are clearly displayed in Russia's war against Ukraine. Religious rhetoric can be used either to legitimise the attack or to espouse its repulsion. Shakman Hurd also draws attention to the role of those nonreligious actors in world politics whose activities impact the management of religious issues worldwide (especially the USA, the UN, and the EU; Russia definitely qualifies as such an entity in the Orthodox world) and which can consequently shape religion as well as politics. As she observes, 'They impact lived experiences of religion and shape local negotiations of religious difference. Religious and political lives are transformed in the process' (Shakman Hurd 2012, p. 960). Some IR scholars see an analogy between postsecularism and the development of other IR concepts (e.g., K. E. Brown refers to the concept of gender, Brown 2020b). Others apply a specific social philosophy (e.g., M. Barbato invokes William Connolly's understanding of political change as power politics of becoming, Barbato 2020). Having considered the views of various scholars, it has to be stated that the essence of the postsecular approach to IR studies appears to be 'an acknowledgement of the ways in which both religion and the secular are "formatted" by the other' (Brown 2020a, pp. 277–78).

This is also close to Puczydłowski's approach, which ties in with the philosophy of 'mediation' set out in the prose of the outstanding Polish culture creator and Nobel Prize in Literature candidate, Witold Gombrowicz. Reciprocal mediation concerns two narrative (i.e., hermeneutic and explanatory) matrices that have coexisted in the approach to social reality (and in our case, international reality) since the early modern period, viz. the re-

ligious matrix and the secular matrix. Although Puczydłowski's deliberations are firmly embedded in Western (Catholic and Protestant) culture, where the division between the religious and the secular is deeply determined historically and culturally, the method of understanding social reality that he proposes (i.e., as two hermeneutic matrices mediated in opposition to each other) is applicable to the Orthodox culture as well. First, this is because the division between the sacred and the profane is inherent to every Christian denomination, although it is not equally explicit everywhere. Second, as noted above, Orthodox Christianity has begun to draw up its contemporary social teaching(s), which proves that dialogue between the secular and the religious (which implies an appreciation of the division between these two spheres) is definitely part of Orthodox tradition. This process has been accelerated by the Russian Orthodox Church's support for Moscow's aggression towards Kyiv. As shown, the Orthodox world generally sees the war as illegitimate, which has compelled a religious response from other Churches. This in turn has a lot of potential to influence the secular domain, especially the convictions of the faithful and the politics of Orthodox countries. As a result, the reverse process can also occur, i.e., the faithful and the governments of these countries can, e.g., influence the institutional form of religion (vide the Orthodox Church of Ukraine becoming independent from the Russian Orthodox Church).

Puczydłowski based his deliberations on his observation of the development of two parallel narratives in Western culture: the religious, which was ubiquitous and dominant in the Middle Ages, and the secular, which has grown since the early modern period. He calls them 'parallel but mutually dependent hermeneutic matrices' (Puczydłowski 2017, p. 22). The role of a matrix is to provide a template for separate copies. Although certain alterations are possible within this template, they are all connected to a general framework that 'imprints its character on individual products' (Puczydłowski 2017). Puczydłowski contends that 'A hermeneutic matrix denotes the kind of framework that dictates the possible spectrum of reasoning to its users' (Puczydłowski 2017, p. 23). He stresses that in the modern era, 'religion and secularism have taken over the role of the great hermeneutic matrices that positioned European thinking' (Puczydłowski 2017).

The essence of Puczydłowski's thinking lies in his conviction that these two matrices, although set in opposition to each other, are interlocked and 'mediated'. Since the establishment of a competing narrative, the mediaeval religious one has in a sense been 'spied on' by the secular one, which 'appraises and classifies while making use of completely heteronomous categories' (Puczydłowski 2017, p. 27). In these conditions, 'religion' has no longer been able to maintain its independence and has undergone 'radical change'. This is evidenced by it being forced to concern itself with problems that it does not consider its own (Puczydłowski 2017, p. 29). However, the same thing has happened with 'secularity'. This matrix is also contaminated by the position of the opposing side and has definitely stepped outside its 'problems'. The spying and the spied on are thus mutually dependent and interlinked and have lost their 'original naivety' (Puczydłowski 2017, pp. 27–28). From this, it follows that 'nowadays, where there is no religion, there is no secularity, and (even more striking) where there is no secularity, there is no religion' (Puczydłowski 2017, pp. 30–31).

Puczydłowski stresses that this schema has been operating in Western Europe since the advent of modern times, especially the enlightenment. This was when God's existence—a question that is characteristic of contemporary Western culture—first appeared in the secular matrix. Consequently, the religious narrative is coupled with 'the atheist narrative and secular ideology', and after coming together 'cannot return to its former wording' (Puczydłowski 2017, p. 31). Although early Christian and mediaeval heresies and the reformation similarly posed intellectual challenges to the main (Catholic) religious narrative and triggered a similar mediation mechanism, the fundamental question of whether God existed was not one of them. Puczydłowski contends that they were not a challenge on the scale of the contemporary secular matrix (Puczydłowski 2017).

It is hard not to agree with these thoughts, although it is impossible not to notice a certain gap in this reasoning. This has to do with omitting the moment when Christianity emerged in antiquity as a separate new religion. As Piotr Mazurkiewicz explains, popular religiosity in the Graeco–Roman world was replete with gods that lurked around every corner—in the air, water, and under the ground—and whose malevolent power was a frequent menace. As befits pagan religions, magical thinking predominated. The new religion revolutionised this way of thinking by 'de-divinising' the world and did so despite desperate attempts to breathe life into the existing religion, e.g., by instituting the Roman Imperial Cult (Mazurkiewicz 2004, pp. 1188–89). The sacred were severed from the profane and expressly counterposed to it. These two orders have been discerned and recognised in Christianity from the outset, and they form the basis for the development of two 'mediated hermeneutic matrices'. The story of the creation of the world, i.e., its separation from God, consequently leads to the desacralisation, dedivinisation, and 'disenchantment' (Weber) of the human realm (Mazurkiewicz 2004). Christ, despite comprising a peculiar concatenation of two natures, stressed that although he was King, his 'kingdom is not of this world' (John 18:36 KJV). Nor did he have any aspirations to govern on Earth. The new faith brought a 'desacralisation revolution' in its wake (Tischner 1999, cf. Mazurkiewicz 2004, p. 1189) and became the first 'spies' of the reality it found. However, according to mediation–matrix logic, 'the observers' require 'the observed'—the secular domain, including its political and religious connections, was 'resacralised' in the Middle Ages. It has to be stressed that this process took one course in Eastern Christianity and another in Western Christianity. For the present purposes, it is important to note the differences between the two traditions in their approach to secularisation in the modern period and contemporary times.

## 5. Secularisation in the Orthodox Tradition

It is worth considering the possibility and the appropriateness of applying Puczydłowski's postsecular approach to the traditions of the Eastern Churches. Orthodox Christianity seems to be even more susceptible to the reciprocal 'mediation' of the two hermeneutic matrices. This is due to the predominance of the conviction that the 'entry' of religion into the social, including the political, domain is a form of actualising faith. It should be emphasised, however, that understanding the process of secularisation in Orthodox countries and the inclusion of the Orthodox case in the general (Western) narrative of secularisation remains a challenge. As Fokas shows, Orthodoxy has long been relegated to the margins of secularisation theory. It was either completely ignored by researchers (as in the case of Charles Taylor's *A Secular Age*) or interpreted as an exception to the rule ('immunity to secularisation')—one of the points in the long list of Orthodox peculiarities, aside from its 'organic' connection with politics or the difficulty of reconciling it with human rights or liberalism (David Martin) (Fokas 2012).

The difference in the perception of secularisation between countries with an Orthodox and a Catholic–Protestant majority became even more apparent after the enlargement of the EU and the accession of several Orthodox countries. Cyprus, Romania, and Bulgaria were now added to the problematic 'Greek exception'. The difficulties in implementing certain aspects of EU legislation in these countries not only strengthened confidence in Orthodox 'immunity' to secularisation but became an impetus for interpreting the EU and the West as the main source of secularisation for Orthodox countries. It should be stressed that this line of argumentation is particularly clearly traced in the rhetoric of Russian religious and secular leaders and is one of the most frequently employed attempts to legitimise Russian aggression against Ukraine.

At the same time, as Hovorun (Hovorun 2022a) showed, the processes of secularisation in the Orthodox countries began long before modernity and proceeded in several stages. The first was the loss of the Church's influence on the government, which came with the conquest of Orthodox territories by the Persian, Arab, Mongolian, and Turkish empires. As a result, the church no longer had a 'likeminded' government to deal with,

but one that had other priorities and which was beholden to other religions, the most often to Islam. Faced with this situation, the church's attention gradually shifted from the central government to local communities. These were to open the door to the formation of the Orthodox version of civil religion and would become the foundation for a number of national liberation/emancipation movements (Hovorun 2018).

In this context, although Ukraine stands out from the chronological scale, it is a demonstrative example of such an emancipation dimension. In recent history, every attempt to gain independence has been accompanied by a request for an independent church: (i) the creation of the Ukrainian Autocephalous Orthodox Church (UAOC, 1919) during the Ukrainian War of Independence 1917–1921; (ii) 1990 and 1991 appeals for granting autocephaly to the Ukrainian Exarchate of the Moscow Patriarchate (later the Ukrainian Orthodox Church of the Moscow Patriarchate), the third revival of the UAOC (1990) and the creation of the Kyiv Patriarchate (1992) at the time of the collapse of the USSR; and (iii) a request for a united Orthodox Church of Ukraine (2018–2019) at the time of Ukraine becoming a real actor in the international arena after the beginning of Russian aggression (2014-). This connection was succinctly summarised by President Poroshenko, who compared the Tomos on Autocephaly to the Act of the Declaration of Independence of Ukraine (Poroshenko: Tomos for Ukraine Is Another Act of Declaring Independence 2019).

In the second stage, which was marked by an even more radical break with the church, secularisation was established as a general norm in a number of European countries. One of the unexpected consequences of such a dramatic separation between church and state was the replacement of religion by ideology or the alliance of religion and ideology, which received the name 'political religion' or quasireligion. Bolshevism, fascism, and Nazism became the most prominent examples of this.[20]

Orthodox countries were most affected by the first of these. In general, the communist government managed to create a system that combined the uncombinable elements. On the one hand, despite the formal declaration of freedom of conscience, the government pursued a policy of exterminating religion and religiosity (further secularisation). On the other hand, after the Second World War, it actively used religious organisations as a tool for implementing its foreign policy (Fletcher 1973). The central place in this process went to the Russian Orthodox Church, which was assigned a double political role—rallying the Orthodox abroad and supporting the Soviet government in the international arena.

With the collapse of the Soviet Union, the Russian Orthodox Church remained almost the only real institution that united the territories of the former Soviet Empire. For a while, even the church itself identified its 'canonical territories' as the former USSR. This became one of the reasons for equating the break with the Soviet past with the break with the 'Russian/Soviet' church.

The Orthodox Churches generally had very different experiences of secularisation at the end of the 20th century—from decades of the complete prohibition of religion (the situation of the Albanian Orthodox Church during the rule of Enver Hoxha) to alliances with ideologies, the repression and exclusion of religion from the public sphere, restrictions on international activities, and its instrumentalisation in the international arena. These experiences determined the different levels of involvement of Orthodox Churches in international life and their different abilities to respond to international events. An eloquent example is the first wave of Russian aggression against Ukraine which, like several other Russian wars in Orthodox countries, was deliberately ignored by Orthodox Churches. In this context, the reaction to the new wave of Russian aggression in 2022, although restrained, was unprecedented.

Although secularisation proceeded differently in the Orthodox culture than it did in Roman–Latin culture, it can also be traced in the development of this tradition. As with every other branch of Christianity, Eastern Orthodoxy was initially built on the separation of, and antithesis between, the sacred and the profane. Although the development of this religion bears the deep imprint of Muslim subjugation and the stamp of Eastern despotism, it has never lost the specific Christian trait of this dualism. For this reason, Puczydłowski's

two 'mediated hermeneutic matrices' can be successfully applied to this tradition, although it is necessary to be mindful of the features peculiar to Eastern Orthodoxy. These are discussed in this part of the article. It is also worth complementing Puczydłowski's concepts with the considerations of Christianity in antiquity on the rise as discussed above. This would broaden the research perspective of IR studies to transcend the West, which is one of the fundamental objectives of the postsecular approach.

### 6. Concluding Remarks

Russia's war against Ukraine offers a striking example of how religion can influence politics and illustrates how religious concepts can serve as a foundation for legitimising international military aggression. For this reason, the present authors view this as another turning point in the examination of religion in IR studies. Applying Puczydłowski's two mediated hermeneutic matrices makes it possible to confirm that the war has also had the opposite effect and has profoundly affected the interior life of the Orthodox Church from interpreting and applying its social teaching(s) to managing its administrative reorganisation (changing the affiliation of individual parishes and the status of local churches). This in turn has influenced the secular (including the political) domain and will continue to do so. This domain is associated with, e.g., the influence of local Churches (the faithful and the hierarchy) on national governments. The postsecular approach allows for these transformations to be linked to each other and to the processes observed in the world of politics. The influence of religious actors may well seem to have limited significance. After all, as noted above, despite their reasonably consistent condemnation of the war and their oft-repeated appeals to the secular authorities, senior Orthodox clergy have not been able to prevent or stop it. However, their potential profound long-term influence on international relations cannot easily be ignored. This particularly concerns the role of Russia, but it also applies to Ukraine in shaping the new international order. The Russian Orthodox Church's legitimisation of the invasion has presaged a decline in the authority of this centre of Orthodoxy around the world. This legitimisation is tied in with its teaching on war and peace, which similarly holds little appeal. This is all the more so as the faithful have an alternative in the social teaching of the Ecumenical Patriarchate, whose unequivocal stance in the face of Russian aggression may raise their trust.[21] How both Ukrainian Churches fit in the Ecumenical Orthodoxy is not known at present. The war is being played out on many levels, including the religious one, and the new balance of power is going to be contingent on its final outcome in all of them.

The liberal international order that came after the end of the cold war was marked by the dominance of the West under the hegemony of the United States. One important feature was the rise in the significance of religion as an identifying factor that was frequently supplanting secular ideology outside the West. This was poorly explained in mainstream IR studies. Religion was actually treated as a weapon against this Western dominance. It thus seems to have helped disrupt the liberal order in the realm of ideas, thereby leading to the gradual breakdown of the 'Westphalian synthesis' (Philpott 2002). The present conflict has once more forced IR scholars to concede that 'religion matters' cannot be omitted from their studies. This, however, leaves the huge challenge of how to study it. Religion, born on the ruins of this liberal order, will probably be more significant in the new order than it is at present. New IR studies have identified that factors in religion have been elaborated This can be defined as the 'postsecular identity of IR studies,' and it is becoming the sine qua non for the further development of the discipline (Kulska and Solarz 2021).

**Author Contributions:** Conceptualization, A.M.S. and I.K.; methodology, A.M.S. and I.K.; investigation, A.M.S. and I.K.; writing—original draft preparation, A.M.S. and I.K.; writing—review and editing, A.M.S. and I.K.; general coordination, A.M.S. All authors have read and agreed to the published version of the manuscript.

**Funding:** This research received no external funding.

**Institutional Review Board Statement:** Not applicable.

**Informed Consent Statement:** Not applicable.

**Data Availability Statement:** Not applicable.

**Conflicts of Interest:** The authors declare no conflict of interest.

## Notes

1 Those textbooks edited by J. Haynes (2007, 2021) are especially noteworthy.

2 This status is also ascribed to Russia by S. P. Huntington in his vision of post-cold-war international relations (cf. Huntington 1996). It should be noted that Huntington's conception of Orthodox civilisation was one of his most criticised on account of his ignorance of the complicated relations and centres of authority within Ecumenical Orthodoxy, the systematic reduction of the Orthodox Church either to the Slavic countries (in a 1993 article, he even occasionally identifies it as Slavic-Orthodox civilization) and/or the post-Soviet countries, his questionable choice of countries (Kazakhstan as a part of the claimed 'Orthodox civilization'), his minimisation of the historical experiences of belonging to other neighbouring civilisations on the part of Orthodox-majority countries, etc.

3 At first glance, such a humorous classification reflects several aspects. First, the assessment of the content and tone of the appeals and statements of the Orthodox churches and their leaders regarding this war (whether they contain only traditional general appeals for peace or whether they make an attempt to name and condemn the unlawful act and the those who commit it, as required by the social teaching of the Orthodox Church)—generic vs. strident. Secondly, they reflect the assessment of the reactions of the churches in the context of their previous attitude towards the policy of the respective states at the first stage of the war (2014–2022). This is the case of the Russian Orthodox Church, which attempted to justify the war (ridiculous), and the Ukrainian Orthodox Church, which often directly or indirectly supported the Russian war narrative but condemned the Russian invasion on the first day of the full-scale invasion (surprising).

4 For more on the doctrine, its stages of development, and the role in this war, see Hovorun (2022c); Kalaitzidis (2022).

5 For example, see Harned (2022).

6 For example, see Sauer (2022).

7 The Legion is not mentioned directly, but the names match those reported by the Ukrainian authorities (სრულიად საქართველოს კათოლიკოს-პატრიარქის სამდივნო (10 June 2022) [Condolences of the Catholicos-Patriarch of All Georgia; 10 June 2022] 2022; სრულიად საქართველოს კათოლიკოს-პატრიარქის სამდივნო (18 April 2022) [Condolences of the Catholicos-Patriarch of All Georgia; 18 April 2022] 2022).

8 This letter was very negatively received by the Polish public, which overwhelmingly supports Ukraine in the present conflict. It was described as 'shameful' and there were some calls for Sawa to be excluded from ecumenical dialogue. Cf. (Abp Sawa 2023).

9 The letter was published in the Polish media and is available at Zwierzchnik (2023).

10 Namely, the Russian, Antiochian, Georgian, and Bulgarian Churches.

11 The lion's share of the document is dedicated to issues that, one way or another, are related to the state. The document contains chapters on the nation, state, secular law, politics, work, property, crime and punishment, 'personal, family and public morality,' 'the health of the individual and nation,' bioethics, ecology, and the media.

12 When referencing the BSC, we identify in brackets only the perspective chapter (Roman numeral) and perspective article (Arabic numeral) after it.

13 Through debates on Jihad and the Crusades.

14 Even though the Orthodox Church does not share a JWT in the Western sense, a number of studies on the Orthodox 'pecspectives', 'reflections', or 'observations' of this phenomena appeared in recent years. For example, see Asfaw et al. (2012); Karras and Hamalis (2017).

15 For more on the complexity and ambivalence of assessing war, the effects of war on humans, and the moral trauma of war, see Papanikolaou (2017).

16 For example, see (Основы социальной концепции иудаизма в России [Basics of the Social Concept of Judaism in Russia] 2003; Основы социальной концепции Российского объединенного союза христиан веры евангельской [Basics of the Social Concept of the Russian United Union of Christians of the Evangelical Faith] 2002); etc. The documents are united not only by a similar structure (sections on nation, state, ethics and secular law, politics, property, media, 'health of nation', war, and morality) but also partly by a common text (verbatim repetition of some sentences and paragraphs), including in the section on war.

17 Simion identifies a number of factors that influenced the difference in the assessment of the justified use of force in war (and therefore war as such), to which he attributes the system of state–church relations, canon law's ambivalence on the use of force, legislative jurisdiction, Slavic cultural influence and influences of the law of jihād, the peculiarities of the understanding of nationalism and patriotism, etc. (Simion 2015).

18 This legitimation has adopted various forms over time—from the creation of a programmatic manifesto, which made the ideological basis of the invasion (2009 Speech at Russian World Assembly), to the defence of the Russian narrative of the war on the international political and religious stage. See Kirill (2011); His Holiness Patriarch Kirill Addresses the U.N., the European

Council, and the OSCE Concerning Facts of Persecution Against The Ukrainian Orthodox Church in Besieged Southeast Ukraine (2014); Response by H.H. Patriarch Kirill of Moscow to Rev. Prof. Dr Ioan Sauca (2022).

[19]  The thesis needs further research, but several facts might indicate the decline in the soft power of both structures—the ROC and the Russian state. In particular, (1) none of the local Orthodox churches supported the ROC's arguments for the legitimisation of the Russian aggression against Ukraine; (2) none of them joined the ROC-initiated breaks of Eucharistic communication with the Ecumenical Patriarchate or with other local churches that recognised the autocephaly of the Orthodox Church of Ukraine; and (3) many Orthodox theologians condemned the 'Russkij Mir' ideology, promoted by the ROC (Gallaher and Kalaitzidis 2022). According to the Global Soft Power Index, for the last year Russia shifted in ranking from 23rd to 105th place (Russia Has Lost Soft Power War with Ukraine—Global Soft Power Index 2023 2023). At the same time, the connection between the decline in the influence of the church and the state needs deeper study, as well as the specifics of this process in the countries with an Orthodox majority, where Russia has successfully positioned itself as a protector of the Orthodox faith (e.g., Religious Belief and National Belonging in Central and Eastern Europe 2017, pp. 14–17).

[20]  For more on the connection between secularisation and civil and political religions, as well as their deployment in Orthodox countries, see Hovorun (2018, chp. 1–2).

[21]  At the same time, this increased raise in trust does not mean that local churches will cease to compete or that their interests will cease to differ. A case in point is the recognition of the autocephaly of the Archdiocese of Ohrid (North Macedonia) by some local churches (Serbian, Bulgarian, Romanian), despite the lack of such recognition from Constantinople.

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
