# Peer review of "The Reactions of Orthodox Churches to Russia’s Aggression towards Ukraine in the Light of the Postsecular Approach to IR Studies"

_religions, doi:10.3390/rel14040515_

Round 1

Reviewer 1 Report

Dear Authors,

the paper is very interesting and actual!

I have some comments regarding chapter 4., "Attitudes of Orthodox Churches towards war...." This is a lengthy and complex chapter that connects various strands of thought, some theoretical and some based on examples from the current war situation. It could be split into two or more parts. However, this is not the main point. There are also some issues with argumentation. With two notable exceptions, the argumentation proceeds convincingly in general. The first is in lines 492-494, where the why "that the waning of the Russian Orthodox Church's authority will lead to the waning of Moscow's authority among Orthodox countries" should be more elaborated.

The second exception appears in the final paragraph, beginning with line 628. The presentation of Antiquity's religions is not accurate; on the contrary, it is quite superficial. You rely solely on Mazurkiewicz's article, which may contain errors of its own. One striking issue is that you refer to Late Antiquity religions as "archaic religions" (line 633). The Roman Imperial Cult, on the other hand, was not a "attempt to breathe life into existing religiosity" (line 635), but served primarily political purposes. It is also worth noting that Late Antiquity is regarded as a period of secularization, which meant that Christianity had to fight not only against pagan religions, but also against a secularized mindset. Another aspect is that you write about a resacralisation of the world in the Middle Ages, after the "desacralization" of the human realm operated by Christianity in the Antiquity. You have no reference for this idea regarding the Middle Ages... It is not appropriate to expedite ideas that are critical to your argument in such a short and underdeveloped paragraph that relies solely on one source, one that is not even available in languages with a wider distribution. You need a stronger foundation for your ideas - there is plenty of literature on this - as well as better argumentation.

In general, your ideas should be founded on a broader foundation. It is impossible to discuss the "just war" theory in relation to Orthodoxy without mentioning Marian Simion's contributions to the field! See e.g. "Seven Factors of Anbivalence in Defining a Just War Theory in Eastern Christianity", 2008; "Just War Theory and Orthodox Christianity", 2011;  "Religion and political conflict", 2011; "Religion in political conflict", 2012; "War and the Right to Life: Orthodox Christian Perspectives", 2015; The ambivalence of ritual in violence: Orthodox Christian perspectives", 2017. Other authors with contributions in the field of Orthodox Christianity, social engagement and especially war are Pantelis Kalaitzidis and Aristotle Papanikolaou - they remain unmentioned, too.

There are also some other small aspects that need to be improved. E.g. you do not mention when the autocephaly of the OCU was granted. Not every reader knows it. You speak only about its recognition (line 157f.) And you mention the letter of Metropolitan Sawa from 4 February 2023, which refers to a previous "customary personal despatch", without mentioning when this dispatch is dated (line 269). It is important at least to know if it was before or after the beginning of the war.

Well, a demanding study like yours always has aspects that need to be improved. But the article is good and it deserves to be improved as stated above!

Author Response

Dear Reviewers,

Thank you very much for the kind reception of our article and for all the comments. We learned a lot from the reviews, however, we have not been able to apply all the comments to this article, even almost all of them have enriched us greatly and will certainly be our guidelines for the future.

Response to reviewers' suggestions:

Reviewer 1:

  1. <<I have some comments regarding chapter 4., "Attitudes of Orthodox Churches towards war...." This is a lengthy and complex chapter that connects various strands of thought, some theoretical and some based on examples from the current war situation. It could be split into two or more parts. However, this is not the main point.>>

We considered this together with remark no. 6 of Reviewer 2, but in the end we would prefer to leave as it is.

  1. <<There are also some issues with argumentation. With two notable exceptions, the argumentation proceeds convincingly in general.>>
  • <<The first is in lines 492-494, where the why "that the waning of the Russian Orthodox Church's authority will lead to the waning of Moscow's authority among Orthodox countries" should be more elaborated.>>

We added the note (no. 19), listing the facts that can serve as indicators of the processes. At the same time agree with the Reviewer 1, that this statement needs further research, that was also stated in the note.

  1. <<The second exception appears in the final paragraph, beginning with line 628.
  • The presentation of Antiquity's religions is not accurate; on the contrary, it is quite superficial. You rely solely on Mazurkiewicz's article, which may contain errors of its own.

See our answer very beneath.

  • One striking issue is that you refer to Late Antiquity religions as "archaic religions" (line 633).

We have changed for <<pagan religions>>, it is clearer now.

  • The Roman Imperial Cult, on the other hand, was not a "attempt to breathe life into existing religiosity" (line 635), but served primarily political purposes.

In our view, these two things were not necessarily mutually exclusive. However, we changed to the revival of religion, not religiosity, because it is not known how that affected religiosity, the fact is that it added a certain freshness to religion. That's what we're all about here.

  • It is also worth noting that Late Antiquity is regarded as a period of secularization, which meant that Christianity had to fight not only against pagan religions, but also against a secularized mindset.

It seems that in this approach the Reviewer 1 applies our contemporary meaning of secularization to the ancient times. That secularization and the modern one are probably two different things, although I may be wrong as a non-expert in this field. This desecularization in ancient times (which could be close to the desacralization that Christianity brought) prepared the ground for the development of Christianity (Christians were even considered atheists by many of those times). But thank you very much for this observation as I am non-expert.

  • Another aspect is that you write about a resacralisation of the world in the Middle Ages, after the "desacralization" of the human realm operated by Christianity in the Antiquity. You have no reference for this idea regarding the Middle Ages...

 The source is the same as described below.

  • It is not appropriate to expedite ideas that are critical to your argument in such a short and underdeveloped paragraph that relies solely on one source, one that is not even available in languages with a wider distribution. You need a stronger foundation for your ideas - there is plenty of literature on this - as well as better argumentation.>>

In our defense we have that it is only a short piece of our discourse. We do not consider this passage to be more that crucial for the whole article, but thanks very much for this observation. The approach to secularization (as desacralization), which we cite after prof. Piotr Mazurkiewicz (we must mention that he is one of the most outstanding researchers of religion and politics in the Polish academy; we have no reason to consider his interpretation wrong, more possible our summary of it was not correct), may seem quite paradoxical, but it is one of the approaches presented by secularization researchers. Mazurkiewicz refers here to German literature, especially the well-established thought of Friedrich Gogarten, who emphasizes that the roots of secularization lie in the Christian faith itself and in its theology of creation. According to Gogarten, however, the seed of freedom contained in the Gospel was soon smothered by pagan remnants, which resulted in the resacralization of the world, which reached its apogee in the Middle Ages. Moreover, the currently observed phenomenon of secularization is therefore a natural process of purifying the Christian religion of pagan accretions, which means liberating Christianity from foreign influences and returning to its sources (cf. F. Gogarten, Verhängis und Hoffnung der Neuzeit. Die Säkularisierung als theologisches Problem, Stuttgart 1953).

Nevertheless, we are grateful to the Reviewer 1 for this interpretation of our text, it is a very important remark for us for the future - the difference between our idea and its interpretation by the reader.

  1. <<In general, your ideas should be founded on a broader foundation. It is impossible to discuss the "just war" theory in relation to Orthodoxy without mentioning Marian Simion's contributions to the field! See e.g. "Seven Factors of Ambivalence in Defining a Just War Theory in Eastern Christianity", 2008; "Just War Theory and Orthodox Christianity", 2011;  "Religion and political conflict", 2011; "Religion in political conflict", 2012; "War and the Right to Life: Orthodox Christian Perspectives", 2015; The ambivalence of ritual in violence: Orthodox Christian perspectives", 2017. Other authors with contributions in the field of Orthodox Christianity, social engagement and especially war are Pantelis Kalaitzidis and Aristotle Papanikolaou - they remain unmentioned, too.>>

We tried to respond to this remark from two sides. On the one hand, we clarified in annotation and text that we consider the positions of the churches, based primarily on their synodal documents (as it was kindly suggested by Reviewer 2, §3). This reflects the de facto approach in the text. The choice in favour of an emphasis on synodal documents was initially dictated by the need to narrow the text, given the rather broad topic.

On the other hand, we added clarifications that some of the problems raised in our text are considered more widely and in more detail in separate works on those issues by a number of authors, in particular those mentioned by the reviewer. Such clarifications are made mainly through the addition of notes (no 14, 15, 17) that outline the direction of the discussion or argumentation.

  1. <<There are also some other small aspects that need to be improved. E.g. you do not mention when the autocephaly of the OCU was granted. Not every reader knows it. You speak only about its recognition (line 157f.) And you mention the letter of Metropolitan Sawa from 4 February 2023, which refers to a previous "customary personal despatch", without mentioning when this dispatch is dated (line 269). It is important at least to know if it was before or after the beginning of the war.>>

We added the date of granting the autocephaly of the OCU.

The 14th anniversary of the enthronement of Patriarch Kirill is February 1, 2023, so after the full-scale invasion of Russia, carried out almost a year earlier. It is not clear when exactly Metropolitan Sawa's letter with congratulations was sent, it is known that the Moscow Patriarchate posted it on its website along with other congratulations around the anniversary. We added an explanation in the text that the letter was sent a few days before the storm it caused in Poland. Thank you for your insightful observations.

Reviewer 2 Report

The article is well-researched, and insightful piece that delves into the complex interplay between religion, politics, and international relations. The authors present a thorough examination of the responses from Orthodox churches, which significantly contributes to the current discourse on the role of religion in global politics. The authors decision to adopt a postsecular approach to IR studies sets the stage for a refreshing perspective on the issue.

By challenging the conventional secularist assumptions prevalent in the field, the article invites the reader to consider the crucial role that religion continues to play in contemporary society, even in the face of modernization and secularization.

The in-depth analysis of the reactions of Orthodox churches, including the Russian Orthodox Church, the Ukrainian Orthodox Church, and other autocephalous Orthodox churches, demonstrates the authors' extensive knowledge and thorough research. The article showcases a range of responses from these churches, reflecting the diverse and complex landscape of Orthodox Christianity.

The nuanced presentation of these reactions avoids oversimplification and provides a rich understanding of the various factors at play. Moreover, the article's organization is commendable, with clear headings and a logical flow of ideas. The writing is engaging and accessible, making complex theories and concepts easily digestible for readers from various backgrounds.

Nonetheless, the following improvements can be made: 1. The article of George Demacopoulos is not included in the bibliography; 2. Rows 100-114 are taken from the abstract and they are a clear repetition; 3. It is important to emphasize in the abstract (rows 15-16) the fact that the article analyses the social teachings of the ROC and EP on the subject of war according to their official synodal documents. 4. Avoid using ”Orthodox social doctrine” (row 288) because the Orthodox Church has no social doctrine, as the Catholic Church has, especially in the context of analyzing two texts of two different Orthodox Autocephalous Churches, that have to different social concepts or teachings. 5. Avoid using ”Panorthodox” (row 387) because footnote 10 clearly shows that it was not a Pan-Orthodox Council. The official title was ”The Holy and Great Council”. 6. In the article under review, the first part of Chapter 4 delves into MiÅ‚osz PuczydÅ‚owski's analytical approach to international relations and his hermeneutical matrices. While the content itself is enlightening and intellectually stimulating, it might be beneficial to address this methodological approach at the end of the first chapter to enhance the article's overall structure and coherence. By relocating this methodological discussion, the article would gain a more solid foundation from the outset, ensuring a seamless transition between chapters and avoiding the potential confusion of an introductory section appearing in the middle of the paper. This restructuring would allow readers to fully appreciate the depth and impact of PuczydÅ‚owski's hermeneutical matrices and the analytical approach to international relations throughout the entire article.

Despite this minor structural concern, the paper remains a commendable and insightful exploration of the complex interplay between religion and geopolitics in the context of Orthodox churches' reactions to Russia's aggression towards Ukraine.

Author Response

Dear Reviewers,

Thank you very much for the kind reception of our article and for all the comments. We learned a lot from the reviews, however, we have not been able to apply all the comments to this article, even almost all of them have enriched us greatly and will certainly be our guidelines for the future.

Response to reviewers' suggestions:

<<The following improvements can be made:

  1. The article of George Demacopoulos is not included in the bibliography;>>

Thank you for noticing that. Done

  1. <<Rows 100-114 are taken from the abstract and they are a clear repetition; >>

We have change the abstract a bit but it is difficult to put it in better words than words taken out of the text.

  1. <<It is important to emphasize in the abstract (rows 15-16) the fact that the article analyses the social teachings of the ROC and EP on the subject of war according to their official synodal documents. >>

Thank you so much for this recommendation/tip. Done

  1. <<Avoid using ”Orthodox social doctrine” (row 288) because the Orthodox Church has no social doctrine, as the Catholic Church has, especially in the context of analyzing two texts of two different Orthodox Autocephalous Churches, that have to different social concepts or teachings.>>

Done. We changed term “doctrine” to concepts or teaching(s), depending on which of the last two options was more appropriate in the sentence.

  1. <<Avoid using ”Panorthodox” (row 387) because footnote 10 clearly shows that it was not a Pan-Orthodox Council. The official title was ”The Holy and Great Council”. >>

Done, the term ”Panorthodox” was changed to ”The Holy and Great Council”.

  1. <<In the article under review, the first part of Chapter 4 delves into Miłosz Puczydłowski's analytical approach to international relations and his hermeneutical matrices. While the content itself is enlightening and intellectually stimulating, it might be beneficial to address this methodological approach at the end of the first chapter to enhance the article's overall structure and coherence. By relocating this methodological discussion, the article would gain a more solid foundation from the outset, ensuring a seamless transition between chapters and avoiding the potential confusion of an introductory section appearing in the middle of the paper. This restructuring would allow readers to fully appreciate the depth and impact of Puczydłowski's hermeneutical matrices and the analytical approach to international the entire article.>>

This point is very valuable and has been considered for a longer time. It corresponds to the comment of Reviewer 1 that part 4 of the article "could be split into two or more parts". Ultimately, however, we decided to leave the structure as it is now, being aware of the inconvenience for the reader. Puczydłowski's matrices are not used in the analysis of the attitudes of the Orthodox Churches towards the war in Ukraine. They are necessary to draw conclusions for the possible consequences of these attitudes for international relations. This chapter is closely related to the last one concerning secularization processes in the Orthodox tradition. The transfer of considerations about matrices had to be associated with the decomposition of the entire article. Of course, that was taken into account as well. However, we feel that this current arrangement better serves the purpose of the text. We analyze the reactions of the Churches from the point of view of religious studies and then include considerations aimed at explaining the consequences of theological analysis into IR study.

Reviewer 3 Report

The artcle discusses the role of religion, particularly the Russian Orthodox Church, in the context of the Russian-Ukrainian conflict. It explores how the church's position has evolved through different stages of the conflict and how it has impacted the administrative structure of the church, leading to demands for autocephaly and the formation of a unified Orthodox Church of Ukraine (OCU).

The text also examines the responses from various Orthodox Churches to the conflict, dividing them into two groups: the generic (Bulgarian, Serbian, Jerusalem, and Georgian Churches) and the strident (Ecumenical Patriarchate, Greek, and Romanian Churches). The primary difference between these groups is the strength of their rhetoric and their willingness to call for peace, as well as their ability to give an assessment of the conflict, including identifying the parties involved and determining the nature of the conflict as aggressive or defensive.

In conclusion, the article highlights how Orthodox Churches condemn war as a tool for solving problems but also emphasize the illegitimacy of the military operations involved in the Russian-Ukrainian conflict.

Author Response

Dear Reviewers,

Thank you very much for the kind reception of our article and for all the comments. We learned a lot from the reviews, however, we have not been able to apply all the comments to this article, even almost all of them have enriched us greatly and will certainly be our guidelines for the future.

Response to reviewers' suggestions:

  1. <<The article discusses the role of religion, particularly the Russian Orthodox Church, in the context of the Russian-Ukrainian conflict. It explores how the church's position has evolved through different stages of the conflict and how it has impacted the administrative structure of the church, leading to demands for autocephaly and the formation of a unified Orthodox Church of Ukraine (OCU).The text also examines the responses from various Orthodox Churches to the conflict, dividing them into two groups: the generic (Bulgarian, Serbian, Jerusalem, and Georgian Churches) and the strident (Ecumenical Patriarchate, Greek, and Romanian Churches). The primary difference between these groups is the strength of their rhetoric and their willingness to call for peace, as well as their ability to give an assessment of the conflict, including identifying the parties involved and determining the nature of the conflict as aggressive or defensive. In conclusion, the article highlights how Orthodox Churches condemn war as a tool for solving problems but also emphasize the illegitimacy of the military operations involved in the Russian-Ukrainian conflict.>>

We are very grateful to your kind remarks. Since the review does not include recommendations for changes, we made no changes to the text in relation to this review.

Thank you all again for your kind and valuable feedback.